# Why Does SARS-CoV-2 Infection Induce Autoantibody Production?

**DOI:** 10.3390/pathogens10030380

**Published:** 2021-03-22

**Authors:** Ales Macela, Klara Kubelkova

**Affiliations:** Department of Molecular Pathology and Biology, Faculty of Military Health Sciences, University of Defence, Trebesska 1575, 500 01 Hradec Kralove, Czech Republic; ales.macela@unob.cz

**Keywords:** COVID-19, SARS-CoV-2, autoantibodies, innate immune recognition, ACE2 signaling

## Abstract

SARS-CoV-2 infection induces the production of autoantibodies, which is significantly associated with complications during hospitalization and a more severe prognosis in COVID-19 patients. Such a response of the patient’s immune system may reflect (1) the dysregulation of the immune response or (2) it may be an attempt to regulate itself in situations where the non-infectious self poses a greater threat than the infectious non-self. Of significance may be the primary virus-host cell interaction where the surface-bound ACE2 ectoenzyme plays a critical role. Here, we present a brief analysis of recent findings concerning the immune recognition of SARS-CoV-2, which, we believe, favors the second possibility as the underlying reason for the production of autoantibodies during COVID-19.

The ongoing COVID-19 pandemic reopens the question of why infections induce the production of natural antibodies (NAbs) having the character of autoantibodies (AAbs). These originally were referred to as natural AAbs. According to clinical data, patients infected with severe acute respiratory syndrome coronavirus 2 (SARS-CoV-2) have a limited spectrum of AAbs specificities, among them anti-phospholipid, anti-interferon alpha and omega (both are type I interferons), anti-interleukins, anti-chemokines, anti-52 kDa SSA/Ro and 60 kDa SSA/Ro ribonucleoproteins, and anti-cardiolipin AAbs [1,2,3,4,5,6]. The presence and level of AAbs frequently detected in patients with COVID-19 are significantly associated with complications during hospitalization and more severe prognoses [7]. The authors of this contribution note that association of AAbs with an unfavorable prognosis possibly reflects a pathogenetic role of immune dysregulation [7]. The arguments for this assertion were based on the clinical data of COVID-19 patients, spectrum of autoantibody specificities, and the correlation between autoantibody-positive patients vs. autoantibody-negative patients that are presented by Pascolini et al. [7]. An analogous argument was also given in the publication by Tay et al. [8].

It is generally accepted, however, that the production of NAbs targeting autoantigens in healthy individuals is an evolutionarily fixed natural process that first arose in cartilaginous fish. Their pentameric IgM is thought to act as an innate-like, T-independent first-line defender until an antigen-specific response can be developed [9].

NAbs having the character of AAbs have been studied since the 1940s [10,11,12,13], and their name was chosen because they are produced at birth in the absence of exposure to foreign antigens. The main characteristics of NAbs, the majority of which are of the IgM isotype, are polyreactive with low binding affinity but high avidity. It became clear very early, however, that, along with the spontaneous production of NAbs, infections of animals by microbial agents induce the production of antibodies reacting with the host’s molecular components [14,15,16]. This situation, in humans, is clearly demonstrated following SARS-CoV-2 infection.

We have tested the production of NAbs using a murine model during very early infection with virulent *Francisella tularensis* subsp. *holarctica*, strain FSC 200 [17]. The majority of such infections induced the production of antibody clones during 12, 24, and 48 h post-infection, reacting with bacterial proteins having orthologs or analogs in eukaryotic cells. These were predominantly of the IgM isotype, but IgG3 and IgA isotypes were also identified. The production kinetics and half-life in the sera of infected mice varied for individual antibody specificities. Generally, we can state that the production of individual antibody specificities during very early intervals post-infection (up to 48 h) were temporary. The composition of the antibody clones was specific at all intervals tested. Some of the specificities produced during this innate, T-independent phase of immune response were already detected during the adaptive phase of immune response to *F. tularensis* infection [18,19,20,21,22]. Thus, the characteristics of such a humoral response to infection allowed us to denominate these early produced antibodies as infection-induced NAbs, some of which are autoreactive.

We therefore assume that NAbs having the character of AAbs can be divided into three groups according to their origins and kinetics. The first consists of NAbs originating from B1 cells located in the spleen, and possibly in the bone marrow, and their production is independent of the presence of gut microbiota. These cells constitute the largest number of spontaneously IgM-secreting cells under so-called naïve conditions. The NAbs produced by these B1 cells represent real, non-antigen-induced NAbs targeting housekeeping functions, including the binding of entities expressing dominant molecular microbial traits, such as lipopolysaccharide. The second group consists of NAbs produced by B1a cells in pleural and peritoneal cavities and characterized as responder cells [23]. We denoted these as infection-induced NAbs. These NAbs represent the real first line of defense, which can be associated with the needs of the classical complement activation pathway. Cooperation between NAbs and complement system components is needed, in some situations, for the internalization of microbes into phagocytes acting as antigen-presenting cells, and it is crucial for deciding the pathogen’s intracellular fate, and, subsequently, to creating signals for the induction of adaptive immune response, as in the case of our *Francisella* model [24,25,26]. The third group of NAbs, targeting autoantigens, is produced during the phase of adaptive immunity on the basis of so-called trained immunity, which was defined as innate immune memory [27,28,29]. Regulation of immune processes at the phase of the adaptive immune response seems to be the dominant role of this third group of NAbs. The incorrectly processed induction and regulation of adaptive immunity by innate immune mechanisms can contribute to a chronic hyper-inflammatory state or the inability to maintain homeostasis, both of which may result in tissue damage and organ failure. Although the human B1 cells and their subtypes are not precisely defined phenotypically [30,31], it is likely that the model of NAbs categorization described above can be, with certain probability, applied to humans. In the literature, the clinical severity of COVID-19 and the presence of serum AAbs are generally thought to be related as a consequence of immune dysregulation. Let us nevertheless take a closer look at this issue.

Angiotensin-converting enzyme 2 (ACE2) has been identified as the receptor for SARS-CoV-2, and it is vital for the viral entry into the host cells [32]. Initial interaction between viral S-protein and extracellular domains of the transmembrane ACE2 proteins counteracts the conversion of angiotensin II (Ang2) to angiotensin 1–7 (Ang1–7), which opposes the action of Ang1–7 realized through MasR, a G protein-coupled receptor. This leads to an increase of Ang2 level and shifts anti-inflammatory action of Ang1–7 to proinflammatory response brought about by Ang2 at the AT1 receptor. At this point, however, internalization of the ACE2-SARS-CoV-2 complex and activation of AT1 receptor by an elevated level of Ang2 initiates increased activity of ADAM17 (disintegrin and metalloproteinase 17), which mediates the proteolytic cleavage of surface ACE2 and enables it to counteract the Ang2 proinflammatory action [33,34,35]. ACE2, as an ectoenzyme, signals, using its circulating peptide targets, components of the renin-angiotensin signaling (RES) pathway. Misregulation of RES might be the reason why, during early stages of infection, SARS-CoV-2 behaves as if it is invisible to the innate immune system and initiates pathophysiological changes in the host tissues. The proinflammatory activity of Ang2 is certainly projected into the bystander cells, which are not infected by the virus, and, in such a manner, are also “instructed” to participate in the proinflammatory action together with the infected cells themselves. This signaling scheme might prolong the virus recognition, on the one hand, and on the other hand, intensify the innate response of cells inside the tissues. The relatively long lag phase enables establishing an innate aggressive inflammatory response known as cytokine storm [36,37,38] with the presence of serum AAbs in severe and critical cases of COVID-19 [4]. Clinically, these patients have acute respiratory distress syndrome, frequently acute cardiac injury, acute kidney injury, and even multi-organ dysfunction with such common complications as coagulopathy and thrombocytopenia [39].

The interaction of SARS-CoV-2 with the ACE2 receptor using the extracellular signaling pathways might be a source of the virus stealth phenotype. The recognition of the virus by ACE2 does not necessarily constitute real innate immune recognition, which might be realized by cytosolic pattern recognition receptors [40]. The SARS-CoV-2 genomic and subgenomic transcripts have been identified in endoplasmic reticulum membranes, mitochondrial membranes and matrix, and in nucleolus, where they potentially hijack the host cell’s machinery and modulate the activation of the host’s cell signaling pathways [41,42,43]. The SARS-CoV-2 open reading frames’ manipulation of mitochondria can induce the release of mitochondrial DNA into cytosol, activate the cytosolic pattern recognition receptors and NLRP3 inflammasome [44], or initiate cell damage by oxidative stress [45,46]. Furthermore, as a consequence of SARS-CoV-2’s intensive replication in infected cells and manipulation of their functional and phenotypic potential, the infected cells die and release their molecular components into the surrounding tissue. Finally, between the SARS-CoV-2 and human proteins, there are some protein epitope similarities, so-called molecular mimicry [47,48,49]. However, the immune response to proteins having epitopes common to both SARS-CoV-2 and human proteins is, according to current knowledge, more a matter of activated T cells rather than a T-cell independent B cell response [50]. All these events together create a deadly cocktail of signals for the immune system.

Molecular and functional characteristics of interactions between SARS-CoV-2 and the host cell generate significant immunogenic signals known as Danger-associated molecular patterns that originate from our cells along with the Pathogen-associated molecular patterns of the virus. Both types of signals initiate the innate immune recognition and activation of immune responses. Delaying the innate recognition of the viral immunogenic signals during the lag phase probably leads to accumulation of self-immunogenic signals. In such a case, a rearrangement is likely to occur, where the noninfectious self dominates over infectious non-self, which seems to be less dangerous. In addition, there is certainly a progression of a strong inflammatory response in host tissues, which causes an increased need for regulation [51]. The data regarding the spectrum of AAbs in the sera of COVID-19 patients substantially correspond to this scheme. Antibodies against cardiolipin, which is an important component of the mitochondrial inner membrane, and AAbs to phospholipids, which constitute a key structural component of cell membranes, suggest that these AAbs play a role in housekeeping. Corresponding to the anti-cardiolipin antibodies, the identified anti-β2-glycoprotein 1, which is a multifunctional plasma protein binding cardiolipin, may confirm such a role for these AAbs in eliminating unwanted protein complexes. The antibodies against 52 kDa SSA/Ro and 60 kDa SSA/Ro ribonucleoproteins located in cytosol and nucleus, respectively, and antibodies against MDA5 (melanoma differentiation-associated protein 5)—one of the intracellular pattern recognition receptors, which in naïve situations interacts with the adaptor MAVS (mitochondrial antiviral signaling protein) and directly initiates transcription of the type I interferon genes—might be oriented to the complex of these intracellular proteins with viral RNA and further support the assignment of housekeeping functions to this group of AAbs. A regulatory function for AAbs against MDA5 is rather unlikely because the mechanism for antibody transition across the intact plasma membrane of the target cell is not yet known; their presence, rather, illustrates the housekeeping role of induced AAbs oriented to self-targets originating from damaged cells.

COVID-19 shares a similar inflammatory immune response with autoinflammatory and autoimmune conditions induced by the flare up production of proinflammatory cytokines [52]. Thus, the antibody specificities induced by SARS-CoV-2 infection, having interferon, interleukin, and chemokine targets, are oriented to control the inflammatory response that is a dominant complication of COVID-19. The antibodies might have a different role against type 1 interferons. Type 1 interferons, through either the STAT1/STAT2 and IRF9 signaling pathway, activate the IFN-stimulated genes functionally associated with an antiviral response or act through the homodimer STAT1 or heterodimer CRKL/STAT5 to initiate the transcription of genes controlling immune responses and inflammation [53,54]. Other AAbs can contribute to the control of inflammation by preventing the manifestation of lymphokines and chemokines proinflammatory effect. The IL-6 is a multipotent cytokine with strong proinflammatory orientation, GM-CSF activates at the genome inflammatory program, CXCL1 contributes to the processes of inflammation (through neutrophils activation), CCL2 also contributes to the inflammatory processes by activation of monocyte/macrophage infiltration, CCL15 is expressed only in the lungs by neutrophils and alveolar macrophages, and finally, CCL16 is a chemoattractant for monocytes and lymphocytes. The AAbs against these cytokines have already been detected in the sera of COVID-19 patients [52]. Therefore, we believe these natural AAbs have a regulatory role and as such represent an attempt to harmonize and control activated mechanisms of immune responsiveness. The natural AAbs might also have a regulatory function against MDA5. This protein, with two N terminal caspase activation and recruitment domains, upon activation by viral RNA binding, interacts with the adaptor mitochondrial antiviral signaling protein, which ultimately leads to transcription of the type I interferon genes [55]. In the case of the natural AAbs against MDA5 regulatory function, however, we would have to assume penetration of these antibodies into the cytosol of infected cells, which, according to current knowledge, is impossible under natural conditions.

Such AAbs specificities further support the notion that these autoantibodies are needed for the elimination of self-targets originating from damaged cells. Their role can also be supported by the fact that the new onset of autoantibodies positively correlates with the response to SARS-CoV-2 proteins [6], which suggests a balance between the response to infectious nonself (SARS-CoV-2) and noninfectious self (SARS-CoV-2 damaged cells).

To summarize our opinions as presented here, we hypothesize that the SARS-CoV-2 infection can induce the production of NAbs by interaction with pleural “responding” B1 cells during the lag phase of infection. These cells express the ACE2 receptors [56] and may be infected similarly to lung pneumocytes. The spectrum of NAbs specificities produced is dependent on the individual immune history and on the ontogenetic experience of the human body with microorganisms. The immediate immune status is further modulated by the status of the gut microbiota at the time of infection. Such individual experiences create a basic state of the instructed innate immune system [25,26], which controls, regulates, and enables the expression of all subsequent events of the innate as well as adaptive immune mechanisms. As SARS-CoV-2 replicates inside host cells, the products of the host cell–pathogen interaction are recognized by innate immune sensors and activate the mechanisms of innate immunity. The innate immune response is dominantly dependent on the induction of inflammation, which is a critical step that should be strictly regulated by controlling the inflammation inducers. AAbs constitute an effective regulatory tool that, along with regulatory cytokines, can control inflammation inducers based on products of the virus’s devastating action on host tissues, regardless of whether they are of viral origin or are components or products of the host’s own cells. SARS-CoV-2 infection is complicated by the fact that a considerable number of viral protein epitopes identical to the epitopes of human proteins exist [48,57,58]. If their immune recognition does in fact occur, then this fact may affect the production of AAbs, which can be produced both by memory B cells (trained immunity) and as a result of the adaptive immune response. Thus, molecular mimicry could be a real further, crucial step of the pathogenetic cascade initiated by SARS-CoV-2 infection and a reason for AAbs production. From all that is discussed above, we consider the production of AAbs during SARS-CoV-2 infection as a regulatory process and an attempt to re-establish homeostasis and not as a result of immune processes dysregulation. The production of AAbs seems nevertheless to be a double-edged sword that must be used properly under strict control; otherwise, it can cause severe health complications.

In conclusion, we would like to emphasize that we present this essay on host–pathogen interrelationships with the intent to open up discussions focused on the possible role of innate immune recognition and the subsequent innate immune response during SARS-CoV-2 infection. The character of the first steps of mutual host–pathogen interactions may suggest the final processes of SARS-CoV-2 infection resolution. The analysis of some clinical data suggests that autoantibodies restricting the interaction of SARS-CoV-2 with the host cells expressing ACE2 may lead to some delayed severe complications occurring in affected patients [59,60]. SARS-CoV-2 significantly induces de novo production of autoantibodies [61], but whether they are the result of signals generated by the virus leading to immune disharmony or are an extreme tool to control homeostasis must by clarified by further clinical data and critical bioinformatic analyzes.

## Data Availability

Not applicable.

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
