# Peer review of "Why Does SARS-CoV-2 Infection Induce Autoantibody Production?"

_pathogens, 2021, doi:10.3390/pathogens10030380_

Round 1
Reviewer 1 Report
This revised manuscript presents an analysis to argue that autoantibody production during SARS-CoV2 infection possesses a regulatory role but not dysregulation of the immune responses. This argument is interesting. A reference showing that cells producing NAbs express ACE2 will help a lot. If not, is there a reference showing the connection between pro-inflammatory response by elevated Ang2 and the cells producing NAbs.
Author Response
The authors would like to note that all changes that were suggested were taken into account and adjustments were made in the best manner possible while ensuring accordance with the comments of both reviewers.

Reviewer 2 Report
The concerns have been addressed.
Author Response
All changes were labeled in yellow and two references has been added to the revised manuscript (also Reviewer 1).
This manuscript is a resubmission of an earlier submission. The following is a list of the peer review reports and author responses from that submission.
Round 1
Reviewer 1 Report
The present review of Macela Ales and Kubelkova Klara is unfortunately way beyond my area of expertise. Nevertheless, I have major concerns.
First, the paper derives from the assumption that autoantibodies are a consequence of SARS-CoV-2 infection. For most of the studies, it is unclear whether these autoantibodies existed before or were induced by SARS-CoV2 infection. Moreover, Bastard et al. (Science 2020) showed that autoantibodies against type I interferons were found in patients prior to SARS-CoV2 infection and hypothesized that it was a cause (and not a consequence) of critical COVID-19.
In addition, I don’t understand how the presented elements about the early host-virus interaction and initiation of the immune response provide arguments in favor of a regulatory role of autoantibodies rather than a consequence of immune dysregulation, the two hypotheses of the authors. I found it very speculative and not supported by the current available data.
The authors state line 80 “In the literature, the clinical severity of COVID-19 and the presence of serum AABs are generally thought to be related as a consequence of immune dysregulation”. What are the references? What are the arguments? Why is it not satisfactory? Why a regulatory role of AABs is a more parsimonious explanation? It’s important for me to answer those questions which is not the case here.
Reviewer 2 Report
Ales et al. proposed an interesting hypothesis on the induction of autoantibodies during COVID19. The topic is important and rapidly evolving. However, the authors failed to provide evidence to support the conclusion that the production of AAbs is not the dysregulation of immune responses, instead, the regulation of itself in situations where the non-infectious self poses a greater threat than the infectious non-self. The authors should review the current literature on AAbs induction during SARS-CoV-2 infection and present the supportive data on both hypotheses.
Major concerns:
- The authors did not fully review the recent advances in identifying the AAbs.
https://pubmed.ncbi.nlm.nih.gov/33532787/
https://pubmed.ncbi.nlm.nih.gov/33330894/
https://pubmed.ncbi.nlm.nih.gov/33106819/
https://pubmed.ncbi.nlm.nih.gov/32972996/
- The AAbs against interferon does not suggest the second hypothesis.
- The molecular mimicry does not suggest the second hypothesis. Based on Line 186-187 “Thus, molecular mimicry could be a real further, crucial step of the pathogenetic cascade initiated by SARS-CoV-2 infection and a reason for AAbs production.” , the conclusion “we consider the production of AAbs during SARS-CoV-2 infection as a regulatory process and an attempt to re-establish homeostasis and not as a result of immune processes dysregulation.” cannot be drawn. In another words, the induction of the AAbs is originally against viral proteins, however, since a considerable number of viral protein epitopes identical to the epitopes of human proteins exist, these antibodies are mistakenly targeting human proteins, and therefore led to the induction of autoantibodies. In this case, the AAb production processing is not a regulatory processing, and is nothing to do homeostasis.
Reviewer 3 Report
This manuscript presents an analysis to argue that autoantibody production during SARS-CoV2 infection is not only the dysregulation of the immune response but also possessing a regulatory role. This argument is interesting. However, more references are needed to back up this argument. First, in lines 165-167: [we hypothesize that the SARS-CoV-2 infection can induce the production of NAbs by interaction with pleural “responding” B1 cells during the lag phase of infection. These cells express the ACE2 receptors and can be infected similarly as are lung pneumocytes.]. A reference showing B1 cells with ACE2 is missing. In contrast, in [The human protein atlas], no ACE2 is expressed in B-1 cells (https://www.proteinatlas.org/ENSG00000130234-ACE2/tissue). Second, as mentioned in the manuscript (line 149 and line 163), antibodies could not penetrate into the cells. How can extracellular antibodies regulate the functions of intracellular proteins (e.g., MDA5)? More explanations are needed to clarify these issues.